# Is the Association between Green Tea Consumption and the Number of Remaining Teeth Affected by Social Networks?: A Cross-Sectional Study from the Japan Gerontological Evaluation Study Project

**DOI:** 10.3390/ijerph17062052

**Published:** 2020-03-20

**Authors:** Manami Hoshi, Jun Aida, Taro Kusama, Takafumi Yamamoto, Sakura Kiuchi, Tatsuo Yamamoto, Toshiyuki Ojima, Katsunori Kondo, Ken Osaka

**Affiliations:** 1Department of International and Community Oral Health, Tohoku University Graduate School of Dentistry, Sendai 980-8575, Japan; tohoku.hoshi@gmail.com (M.H.); chester930@gmail.com (T.K.); yamamoto173998@gmail.com (T.Y.); sakurakawamura39@gmail.com (S.K.); osaka@m.tohoku.ac.jp (K.O.); 2Department of Disaster Medicine and Dental Sociology, Graduate School of Dentistry, Kanagawa Dental University, Yokosuka 238-8580, Japan; yamamoto.tatsuo@kdu.ac.jp; 3Department of Community Health and Preventive Medicine, Hamamatsu University School of Medicine, Hamamatsu 431-3192, Japan; ojima@hama-med.ac.jp; 4Center of Preventive Medical Sciences, Chiba University, Chiba 260-0856, Japan; kkondo@chiba-u.jp; 5National Center for Geriatrics and Gerontology, Obu 474-8511, Japan

**Keywords:** green tea, tooth loss, oral health, social network, friends

## Abstract

Consumption of green tea without sugar, as well as social networks, are associated with a lower risk of tooth loss. There is a possibility of confounding both factors because tea is often drunk with friends. Therefore, the present study aimed to examine whether green tea consumption is beneficially associated with the number of remaining teeth, while considering social networks. This cross-sectional study was based on the Japan Gerontological Evaluation Study (JAGES) in 2016. Self-administered questionnaires containing questions about green tea consumption were mailed to 34,567 community-dwelling residents aged ≥ 65 years. We used the number of remaining teeth as a dependent variable, and green tea consumption and the number of friends met over the past month (social network size) as independent variables. Linear regression models with multiple imputation were used. A total of 24,147 people responded (response rate = 69.9%), and 22,278 valid data were included into our analysis. Participants’ mean age was 74.2 years (standard deviation = 6.3), and 45.9% were men. Among the participants, 52.2% had ≥ 20 teeth, 34.2% drank 2–3 cups of green tea per day, and 32.6% met ≥ 10 people over the past month. After adjusting for all potential confounders, both higher green tea consumption and a larger social network size were associated with more remaining teeth (both *p* for trend < 0.001). The association of green tea was greater among those with smaller social networks (*p* for interaction < 0.05). The protective association of green tea was remarkable among people with smaller social networks.

## 1. Introduction

Tooth loss is the leading cause of disability-adjusted life years (DALYs) in oral diseases and conditions, accounting for 7.6 million DALYs [1]. Tooth loss deteriorates both physical and social functions of oral health, such as mastication [2] and oral health-related quality of life functions, including eating, speaking, smiling, sleeping, and communication [3]. In addition to these effects on oral function, tooth loss potentially exacerbates general health status. Previous studies have shown that the number of remaining teeth predicts increases in all-cause mortality, cardiovascular diseases, coronary heart disease [4], and the onset of dementia [5].

Green tea consumption is reported to associate with a lower risk of tooth loss [6,7]. Green tea contains a high quantity of fluoride [8] and polyphenols such as catechins [9], and in most East Asian countries, is drunk often without sugar. Previous studies have shown that it can prevent both dental caries and periodontal disease [10,11,12]. In addition to its beneficial effects on various health outcomes, such as all-cause mortality, cardiovascular disease, cancers, non-alcoholic fatty liver disease, metabolic syndrome, diabetes, and other health outcomes [13,14,15,16,17,18,19,20], green tea consumption is also considered to have an effect on oral health. In spite of abundant benefits from the basic sciences, there are relatively fewer clinical studies on the merit of green tea on oral health [21].

However, previous studies on green tea and health have not considered the effect of the social aspect of green tea consumption. Japan has a traditional tea ceremony culture with over 1000 years of history. In addition to the afternoon tea-drinking culture in England, tea drinking has spread widely into the general population. Because green tea was traditionally drunk with others in the tea ceremony, even now it is often drunk when people meet with others. Therefore, it is possible that the health effects of green tea are confounded by the effect of social relationships [22,23]. Social relationships have beneficial effects on health [24]. In the dental field, social networks are reported to associate with a lower risk of tooth loss [25]. Possible mechanisms between social networks and oral health are stress-buffering and the diffusion of new information and health behaviors through social networks [26,27]. In fact, studies have suggested that smoking behavior and food intake were diffused throughout social networks [28,29,30]. However, no study has examined the association between green tea consumption and the number of remaining teeth with consideration of social networks. Therefore, the present study aimed to examine whether green tea consumption is beneficially associated with the number of remaining teeth, considering social networks.

## 2. Materials and Methods

### 2.1. Ethics Approval

Ethical approval for this study was obtained from the Ethics Committee of the National Center for Geriatrics and Gerontology (approval number: 992) and Chiba University (approval number: 2493). Participants were informed that participation in the present study was voluntary, and that returning the questionnaire with responses was taken as consent to participate.

### 2.2. Study Design, Setting, and Participants

This cross-sectional study investigated whether the association between green tea consumption and the number of remaining teeth was affected by social networks. We analyzed cross-sectional data from the Japan Gerontological Evaluation Study (JAGES), particularly from the 2016 JAGES survey. JAGES is one of the few population-based gerontological surveys in Japan that focuses on social determinants of health and the social environment [31].

Between October and December 2016 (Takahama town was included in the survey only from January 2017 on), self-administered questionnaires were mailed to 279,661 functionally independent community-dwelling residents aged ≥65 years living in 39 municipalities in Japan. The questionnaires containing questions about green tea consumption were mailed to 34,567 residents.

### 2.3. Dependent Variable

We used the number of remaining teeth as a dependent variable. Participants were asked the following question: “How many remaining teeth do you have, including teeth covered by crowns? The total number of remaining teeth on an adult, including wisdom teeth, should be 32.” Respondents chose one of the following options: 0, 1–4, 5–9, 10–19, or ≥ 20 teeth. Following a previous study, we changed these categories into a continuous variable: a score of 0 (complete edentulism), 2.5 (1–4 teeth), 7 (5–9 teeth), 14.5 (10–19 teeth), or 26 (≥20 teeth) [32]. We used this continuous variable as the dependent variable in linear regression analysis.

### 2.4. Independent Variables

We used social network size and green tea consumption as independent variables. We assessed social network size with the following question: “How many friends/acquaintances have you seen over the past month? Count the same person as one, no matter how many times you have seen him/her.” Respondents chose one of the following options: 0, 1–2, 3–5, 6–9, or ≥10 people. We assessed green tea consumption by asking about the number of cups of green tea drunk in a day, categorized as follows: <1, 1, 2–3, or ≥4 cups/day. We used these categorical variables as independent variables in linear regression analysis. We did not include the variables on dental diseases, because they were considered as the mediators between green tea consumption and the number of remaining teeth. If mediators are adjusted, it causes a biased estimation of the association of the main exposure, which is described in the Strengthening the Reporting of Observational Studies in Epidemiology (STROBE) statement as follows: *“Inappropriate decisions may introduce bias, for example, by including variables that are in the causal pathway between exposure and disease (unless the aim is to assess how much of the effect is carried by the intermediary variable)”* [33].

### 2.5. Covariates

We considered the following variables as potential confounders: sex (male/female), age (65–69, 70–74, 75–79, 80–84, or ≥85 years old), smoking status (never, former, or current), years of education (≤9, 10–12, or ≥13 years), history of diabetes mellitus (yes/no), equivalent annual income (USD < 10,000, 10,000–19,999, 20,000–29,999, 30,000–39,999, or ≥40,000; JPY 100 = USD 1), frequency of brushing teeth (≥twice/day or ≤once/day), eating snacks every day (yes/no), and living alone (yes/no).

### 2.6. Statistical Analyses

We applied linear regression models to more easily interpret the results of the interaction. In model 1, social network size, green tea consumption, sex, and age were included. All other variables were added in model 2. In model 3, an interaction term was added to examine the interaction effect between social network size and green tea consumption on the number of remaining teeth. We assumed “missing at random (MAR)” for missing data, and applied multiple imputation by chained equations (MICE) [34]. All analyses were conducted using the Stata 15 software (Stata Corp, College Station, TX, USA).

## 3. Results

Of the target 34,567 population, 24,147 people responded (response rate = 69.9%). For our study, we analyzed 22,278 valid data. Participants’ mean age was 74.2 years (standard deviation = 6.3), and 45.9% were men.

Table 1 shows the descriptive statistics of the respondents. Respondents who had larger social networks and drank more cups of green tea tended to have a higher number of remaining teeth.

Figure 1 shows the association between social network size and green tea consumption (non-responses are excluded). Respondents with larger social networks tended to drink more cups of green tea.

Table 2 shows the results of linear regression analysis with multiple imputation. After adjusting for all covariates in model 2, meeting more than 3 friends/month was significantly associated with a higher number of remaining teeth (model 2). Drinking more than 2 cups of green tea/day was also significantly associated with a higher number of remaining teeth. Overall, larger social network size and higher green tea consumption were significantly associated with a higher number of remaining teeth (both *p* for trend < 0.001). In model 3, there was a significant negative interaction between social network size and green tea consumption (*p* for interaction < 0.05).

Figure 2 shows the interaction effect between social network size and green tea consumption. There was no protective association between green tea consumption and the number of remaining teeth among respondents with larger social network size. The protective association of green tea consumption on remaining teeth was remarkable among people with smaller social network size.

## 4. Discussion

This is the first study to investigate the association between green tea consumption and oral health outcome, with consideration of social networks. The study demonstrated that both higher green tea consumption and meeting more friends per month were significantly associated with having a higher number of remaining teeth. Notably, the protective association of green tea was remarkable among participants with smaller social network size.

The present study’s results for main effects (green tea and social networks on oral health) are similar to previous studies. For green tea consumption, a previous cross-sectional study showed that consumption of more than one cup of green tea per day was significantly associated with a higher number of remaining teeth among 40- to 64-year-old individuals [6]. Another cross-sectional study showed that consumption of green tea 4–5 times per day was significantly associated with a higher number of remaining teeth among pregnant women in Japan [7]. These beneficial associations of green tea consumption on teeth are supported by biological mechanisms through fluoride and catechin [21]. For social networks, a previous cross-sectional study showed that meeting ≥ 1 friend per month was significantly associated with a higher number of remaining teeth among ≥ 65-year-old individuals in Japan [25].

We first hypothesized that these results in previous studies of green tea and oral health were confounded by the effect of social relationships, because Japanese people often drink green tea with others, as is the Japanese tradition. Even after considering social networks, a significant association of green tea consumption with remaining teeth was still observed. However, the present study revealed that the significant association of green tea with teeth was observed among respondents with smaller social network size. There are some possible explanations for this. Participants with larger social network size may gain larger benefits of social support from their friends. Consistent with the results, these participants are considered to have better oral health behaviors. In our data, respondents with larger social networks tended to brush their teeth and visit a dentist more frequently (results not shown). Therefore, it is possible that green tea is less effective for those respondents, who have good oral health behaviors, because of the “ceiling effect.” In contrast, participants with smaller social networks do not gain large benefits from their social networks. Therefore, among these respondents with worse oral health behaviors, the protective effects of green tea on dental caries and periodontal disease were relatively larger. Specifically, the effects of polyphenolic catechins (e.g., epigallocatechin-3-gallate (EGCG), epigallocatechin (EGC), epicatechin-3-gallate (ECG), and epicatechin (EC) [35]) in inhibiting the growth of oral bacteria such as *Streptococcus mutans* [36,37] and *Porphyromonas gingivalis* [38], and of fluoride [8] in facilitating re-mineralization and acid resistance of enamel [39] are considered to reduce the risks of periodontal disease and caries, which are the main causes of tooth loss [40]. These benefits from the green tea could be relatively larger for people with worse oral health behaviors, due to smaller social networks.

This study had several strengths. Firstly, we used a large sample size of 22,278 subjects from 39 municipalities in Japan. Secondly, this was the first study to investigate the effects of green tea on oral health outcomes with consideration of social networks. Thirdly, this was the first study to investigate the association between green tea consumption and the number of remaining teeth among older adults (aged 65 years and older).

This study also had several considerable limitations. Firstly, this study was cross-sectional, and longitudinal studies are required for causal inference. Secondly, this study used self-reported questionnaire data. However, a previous study reported the validity of a self-reported number of remaining teeth [41]. Additionally, because of misclassification by self-reported questionnaire data, the estimated 95% confidence intervals of the association between green tea consumption/social network size, and the number of remaining teeth, are considered to be inflated. Despite this bias, the present study showed significant associations. Therefore, these results are considered to be robust. Thirdly, because the municipalities that participated in the JAGES survey were not selected randomly, the generalizability of these findings to other populations is limited. Fourthly, we did not ask about consumption of beverages other than green tea, and thus could not consider them in the analysis. Finally, we did not measure fluoride exposure. It has been reported that fluoride application was insufficient in Japan [42]. Even in recent years, when fluoride toothpaste has been widely used, an additional fluoride mouth rinse program is still effective in Japan [43]. This historical situation may explain the present beneficial association of green tea with oral health. Therefore, the generalizability of the effects of green tea on tooth loss to countries with water fluoridation or sufficient topical fluoride application is unclear.

## 5. Conclusions

Green tea consumption was positively associated with the number of remaining teeth among older Japanese. The association between green tea consumption and the number of remaining teeth was affected by the social networks. The protective association of green tea consumption on teeth was larger among the people with smaller social network sizes than those with larger network sizes.

## Figures and Tables

**Figure 1 ijerph-17-02052-f001:**
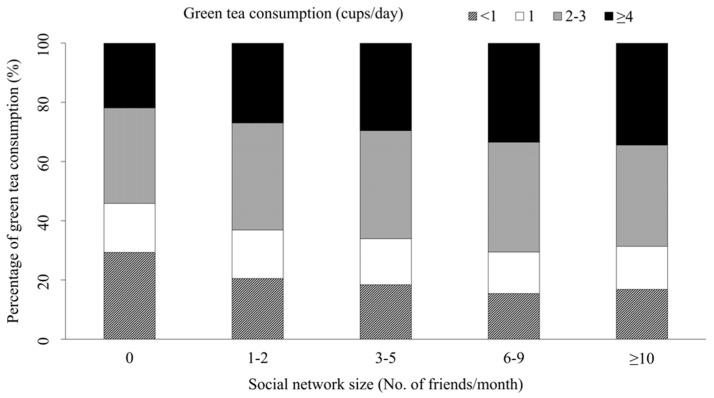
Social network size and green tea consumption.

**Figure 2 ijerph-17-02052-f002:**
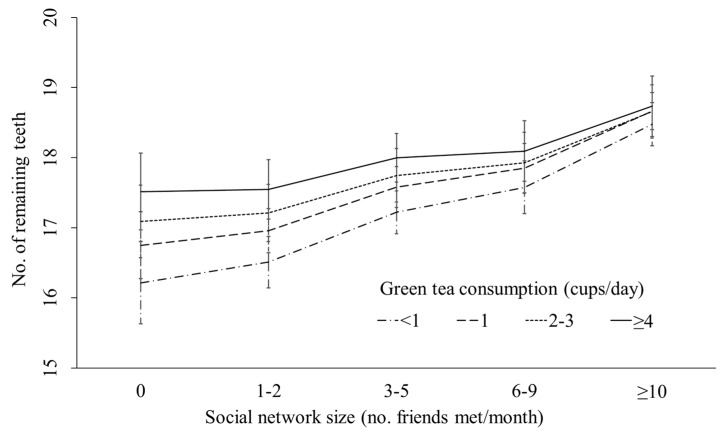
Interaction between green tea consumption and social network size.

**Table 1 ijerph-17-02052-t001:** Descriptive statistics by the number of remaining teeth (N = 22,278).

Characteristics	Subitem	Frequency	No. of Remaining Teeth				
		N	0 (%)	1–4 (%)	5–9 (%)	10–19 (%)	≥20 (%)
**Total**		22,278	8.2	6.1	9.8	20.9	52.2
**Social network size**	0	1865	12.2	8.6	11.1	21.9	43.7
**(no. friends met/month)**	1–2	4097	10.3	7.5	11.0	21.8	46.5
	3–5	5437	7.9	6.5	10.3	22.2	50.6
	6–9	2937	7.3	5.2	9.7	21.7	53.3
	≥10	7269	6.2	4.6	7.8	19.2	60.3
**Green tea consumption**	<1	4027	8.6	7.0	10.1	21.6	50.3
**(cups/day)**	1	3257	7.7	5.5	8.6	22.2	53.7
	2–3	7621	7.9	6.2	9.8	20.9	52.5
	≥4	6596	8.0	5.6	9.6	19.8	54.1
**Sex**	Male	10,222	9.6	7.1	10.2	21.1	50.0
	Female	12,056	7.1	5.4	9.4	20.6	54.1
**Age (years)**	65–69	6568	3.4	3.7	6.8	21.1	63.7
	70–74	5922	6.1	5.2	9.4	21.6	55.8
	75–79	5065	8.7	6.4	11.1	20.6	50.5
	80–84	3169	13.9	9.7	12.2	21.1	37.7
	≥85	1554	23.7	12.3	14.4	17.6	25.0
**Education (years)**	≤9	7406	12.8	8.1	12.5	21.3	41.1
	10–12	8845	6.3	5.5	8.8	21.5	55.8
	≥ 13	5659	4.9	4.4	7.5	19.4	62.2
**Equivalent income**	<10,000	2278	12.0	8.0	14.1	21.8	40.0
**(USD)**	10,000–19,999	6185	8.1	6.2	9.9	21.5	52.1
	20,000–29,999	4161	6.4	4.8	7.8	20.9	58.7
	30,000–39,999	2592	5.3	4.2	6.6	19.8	62.7
	≥40,000	1931	5.3	3.6	7.4	17.8	64.7
**Smoking status**	Never	13,150	6.7	5.0	8.6	19.8	56.9
	Former	6345	10.2	7.2	10.7	21.6	48.4
	Current	2382	11.2	8.7	13.6	25.2	38.7
**Diabetes**	No	18,560	7.9	6.1	9.5	20.7	53.7
	Yes	2782	10.8	6.8	12.0	22.5	45.8
**Brushing teeth**	≥2	15,749	4.5	4.9	9.1	21.2	57.9
**(times/day)**	≤1	5663	16.3	9.3	11.6	20.4	39.5
**Eating snacks every day**	No	7463	6.9	5.5	9.7	19.7	56.3
	Yes	13,748	8.5	6.3	9.6	21.6	51.2
**Living alone**	No	17,249	7.7	5.6	9.0	20.8	54.5
	Yes	2462	8.1	6.8	12.0	21.0	49.6

Non-responses are excluded.

**Table 2 ijerph-17-02052-t002:** Association of each factor with the number of remaining teeth by linear regression analysis with multiple imputation (N = 22,278).

Characteristics	Subitem	Model 1	Model 2	Model 3
		β (95% CI)	β (95% CI)	β (95% CI)
**Social network size**	1–2	0.24 (−0.28 to 0.76)	0.19 (−0.32 to 0.70)	0.38 (−0.14 to 0.90)
**(no. friends met/month) (ref. = 0)**	3–5	1.08 (0.57 to 1.60) ***	0.76 (0.26 to 1.26) **	1.18 (0.55 to 1.80) ***
	6–9	1.57 (1.02 to 2.12) ***	0.95 (0.41 to 1.49) **	1.62 (0.81 to 2.43) ***
	≥10	2.61 (2.12 to 3.11) ***	1.69 (1.21 to 2.18) ***	2.61 (1.62 to 3.59) ***
**Green tea consumption (cups/day)**	1	0.72 (0.28 to 1.16) **	0.35 (−0.08 to 0.77)	0.62 (0.12 to 1.12) *
**(ref. = <1)**	2–3	0.81 (0.45 to 1.18) ***	0.48 (0.13 to 0.83) **	1.05 (0.40 to 1.70) **
	≥4	1.16 (0.78 to 1.53) ***	0.67 (0.31 to 1.04) ***	1.56 (0.64 to 2.48) **
**Sex (ref. = male)**	Female	0.89 (0.64 to 1.15) ***	−1.14 (−1.47 to −0.81) ***	−1.14 (−1.47 to −0.81) ***
**Age (years) (ref. = 65–69)**	70–74	−1.81 (−2.14 to −1.49) ***	−1.53 (−1.85 to −1.22) ***	−1.53 (−1.85 to −1.22) ***
	75–79	−3.13 (−3.47 to −2.79) ***	−2.72 (−3.06 to −2.38) ***	−2.72 (−3.06 to −2.38) ***
	80–84	−5.82 (−6.22 to −5.42) ***	−5.16 (−5.56 to −4.76) ***	−5.16 (−5.57 to −4.76) ***
	≥85	−9.18 (−9.71 to −8.66) ***	−8.17 (−8.69 to −7.66) ***	−8.18 (−8.70 to −7.67) ***
**Smoking status (ref. = never)**	Former		−2.45 (−2.79 to −2.10) ***	−2.45 (−2.80 to −2.11) ***
	Current		−4.33 (−4.78 to −3.88) ***	−4.33 (−4.78 to −3.87) ***
**Education (years) (ref. = ≤9)**	10–12		1.67 (1.37 to 1.96) ***	1.66 (1.37 to 1.96) ***
	≥13		2.45 (2.12 to 2.79) ***	2.46 (2.12 to 2.79) ***
**Diabetes (ref. = no)**	Yes		−0.96 (−1.32 to −0.61) ***	−0.96 (−1.32 to −0.61) ***
**Equivalent income (USD)**	10,000–19,999		1.39 (1.00 to 1.77) ***	1.39 (1.01 to 1.78) ***
**(ref. = <10,000)**	20,000–29,999		1.98 (1.54 to 2.43) ***	1.98 (1.54 to 2.43) ***
	30,000–39,999		2.35 (1.86 to 2.83) ***	2.35 (1.86 to 2.83) ***
	≥40,000		2.60 (2.09 to 3.11) ***	2.60 (2.08 to 3.11) ***
**Brushing teeth (ref. = ≥twice/day)**	≤once/day		−3.50 (−3.80 to −3.20) ***	−3.50 (−3.79 to −3.20) ***
**Eating snacks every day (ref. = no)**	Yes		−1.07 (−1.34 to −0.81) ***	−1.07 (−1.34 to −0.80) ***
**Living alone (ref. = no)**	Yes		−0.27 (−0.65 to 0.12)	−0.26 (−0.65 to 0.12)
**Interaction between green tea consumption and social network size**				−0.09 (−0.17 to −0.002) *

CI: confidence interval; ref.: reference. Model 1: Included social network size, green tea consumption, sex, and age; Model 2: Model 1 + smoking status, years of education, history of diabetes mellitus, equivalent income, frequency of brushing teeth, eating snacks every day, and living alone; Model 3: Model 2 + interaction between green tea consumption and social network size. * *p* < 0.05, ** *p* < 0.01, *** *p* < 0.001.

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
