# Peer review of "Is the Association between Green Tea Consumption and the Number of Remaining Teeth Affected by Social Networks?: A Cross-Sectional Study from the Japan Gerontological Evaluation Study Project"

_ijerph, 2020, doi:10.3390/ijerph17062052_

Round 1
Reviewer 1 Report
The manuscript evaluated the association between the number of remaining teeth and both green tea consumption as well as of social network interaction in a Japanese aged population. Authors adopted a cross-sectional study design and reported that people with both high tea consumption and large social network possessed greater teeth number; with greater association among those with smaller social network.
Manuscript has been improved following revision and resubmission.
Reviewer 2 Report
Many thanks for the revision and incorporating all suggested changes to the manuscript
This manuscript is a resubmission of an earlier submission. The following is a list of the peer review reports and author responses from that submission.
Round 1
Reviewer 1 Report
Article described the possible impact of social network in Japanese aged population upon the linked between consumption of green tea and the number of remaining teeth. Utilizing a cross-sectional study approach, the authors reported that increased intake of green tea and a large social network was associated in higher teeth number in their study population. A few things however need be clarified.
Firstly, the study discussion section requires fine-tuning. Moreover, there are a lot of random numbers scattered across the entire discussion section of the paper; which I presume are references intended to be included by authors (lines 174-176, among others). The actual referenced papers are however missing in the reference section of the manuscript.
Secondly, an important variable missing in the data summaries provided and questionnaire adopted is whether respondents have a prior or current underling dental malady. No mention of this was reported by study authors. What are the authors' opinion of how this variable would have impacted their findings?
Author Response
Thank you very much for reviewing our manuscript and offering valuable advice.
We have addressed your comments with point-by-point responses, and revised the manuscript accordingly.
Please see the attachment.

Reviewer 2 Report
I have reviewed the manuscript “Is the association between green tea consumption and number of remaining teeth affected by social network? : A cross-sectional study from the Japan Gerontological Evaluation Study project” submitted to “Int. J. Environ. Res. Public Health” for publication. I found this work interesting and fit well with in the scope of this journal. Authors have conducted a survey based study considering the association between green tea consumption and number of remaining teeth. The manuscript fits well within the scope of the journal; it needs some major improvements; there are a few suggestions that authors may consider to improve it further:
The use of English language is reasonable, however, there are a number of minor punctuation and grammatical errors; that should be corrected and rephrased using academic English for a better flow of text for reader.
Abstract: covered all key information however, there is not any clear mention of aim of the study in the abstract; please add the precise statement of aim of the study in the abstract.
Introduction; is very detailed and covering the background information and the rationale of the study effectively.
Line 52-53: “green tea consumption is also considered to have an effect on oral health” needs a reference citation; please cite the following reference for this statement.
Khurshid, Zohaib, et al. "Suppl-1, M3: Green Tea (Camellia Sinensis): Chemistry and Oral Health." The open dentistry journal 10 (2016): 166.
Methodology: The study was conducted in 2016-2017; the reason of delay in publication should be justified.
Line 78: seems grammatically incorrect; please check, was there only one question? Please correct and rephrase accordingly.
The section 2.6. Ethics approval should be present in the beginning of methods section
Line 125: Figure 1: there is no need to keep % with each digit
The discussion is reasonable however most of it is not supported by citation of references and a little bit brief; authors must include and cite more studies in context.
The conclusions section has been poorly presented; only two statements have been mentioned that can be further improved;
“The association between green tea consumption and number of remaining teeth was affected by social network.” [please expand this more precisely]
The protective association of green tea consumption on teeth was remarkable among people with a smaller social network size.[what about in general and for larger network size].
Author Response

(The authors gave the same response as above.)
